# Epithelial Cell Adhesion Molecule (*EpCAM*) Expression Can Be Modulated via *NFκB*

**DOI:** 10.3390/biomedicines10112985

**Published:** 2022-11-20

**Authors:** Saadiya Zia, Komal Tehreem, Sidra Batool, Mehreen Ishfaq, Shaher Bano Mirza, Shahrukh Khan, Majed N. Almashjary, Mohannad S. Hazzazi, Husam Qanash, Ahmad Shaikh, Roua S. Baty, Ibrahim Jafri, Nouf H. Alsubhi, Ghadeer I. Alrefaei, Rokayya Sami, Ramla Shahid

**Affiliations:** 1Department of Biosciences, COMSATS University Islamabad (CUI), Islamabad 45550, Pakistan; 2Department of Biochemistry, Faculty of Sciences, University of Agriculture Faisalabad, Faisalabad 38000, Pakistan; 3Research School of Chemistry, Australian National University, Canberra, ACT 2600, Australia; 4Department of Medical Laboratory Sciences, Faculty of Applied Medical Sciences, King Abdul Aziz University, Jeddah 22254, Saudi Arabia; 5Hematology Research Unit, King Fahd Medical Research Center, King Abdul Aziz University, Jeddah 22254, Saudi Arabia; 6Department of Medical Laboratory Science, College of Applied Medical Sciences, University of Ha’il, Hail 55476, Saudi Arabia; 7Molecular Diagnostics and Personalized Therapeutics Unit, University of Ha’il, Hail 55476, Saudi Arabia; 8Department of Clinical Laboratory Sciences, College of Applied Medical Sciences, King Khalid University, P.O. Box 960, Abha 61421, Saudi Arabia; 9Department of Biotechnology, College of Science, Taif University, P.O. Box 11099, Taif 21944, Saudi Arabia; 10Biological Sciences Department, College of Science and Arts, King Abdul Aziz University, Rabigh 21911, Saudi Arabia; 11Department of Biology, College of Science, University of Jeddah, P.O. Box 80327, Jeddah 21589, Saudi Arabia; 12Department of Food Science and Nutrition, College of Sciences, Taif University, P.O. Box 11099, Taif 21944, Saudi Arabia

**Keywords:** epithelial cell adhesion molecule, acute lymphoblastic leukemia, costunolide, cell proliferation, telomerase inhibition

## Abstract

The epithelial cell adhesion molecule (EpCAM) is considered an essential proliferation signature in cancer. In the current research study, qPCR induced expression of *EpCAM* was noted in acute lymphoblastic leukemia (ALL) cases. Costunolide, a sesquiterpene lactone found in crepe ginger and lettuce, is a medicinal herb with anticancer properties. Expression of *EpCAM* and its downstream target genes (*Myc* and *TERT*) wasdownregulated upon treatment with costunolide in Jurkat cells. A significant change in the telomere length of Jurkat cells was not noted at 72 h of costunolide treatment. An in silico study revealed hydrophobic interactions between EpCAM extracellular domain and Myc bHLH with costunolide. Reduced expression of *NFκB*, a transcription factor of *EpCAM*, *Myc,* and *TERT* in costunolide-treated Jurkat cells, suggested that costunolide inhibits gene expression by targeting NFκB and its downstream targets. Overall, the study proposes that costunolide could be a promising therapeutic biomolecule for leukemia.

## 1. Introduction

The epithelial cell adhesion molecule (EpCAM, CD326), a cancer biomarker, is a glycoprotein with an extracellular, transmembrane, and cytoplasmic domain [1,2]. EpCAM has a convincing role in cellular proliferation, differentiation, adhesion, migration, and intracellular signaling [3]. It expresses in embryonic stem cells, cancer stem cells, and germ cells [4]. Over-expression of EpCAM in carcinomas (breast, pancreatic, ovarian, lung, stomach, and gall bladder) is associated with poor prognosis [5]. Lower levels of EpCAM inhibit cell proliferation in breast cancer cells [6]. In acute myeloid leukemia, EpCAM^+^ leukemia cells show augmented chemoresistance and oncogenesis [7]. Upon proteolytic cleavage by γ-secretase, the intracellular domain of EpCAM interacts with the Wnt signaling pathway (β-catenin, FHL2) and translocates to the nucleus, where it regulates the expression of *Myc* as well as *cyclin A* and *E* [8,9]. Myc, along with NFκB and other regulatory proteins (Sp1, AP2), controls telomerase transcription, which is crucial for uncontrolled proliferation of cancer cells [10] (Figure 1).

The potent anticancer properties of sesquiterpene lactones areassociated with the α-methylene-γ-lactone group [11]. Costunolide, a well-known sesquiterpene lactone found in crepe ginger and lettuce, is considered a medicinal herb due to its anticancer properties. It plays a valuable role in the prevention of osteoporosis, diabetes, and ulceration, as well as viral, bacterial, and fungal infections [12]. The drug also has chemopreventive activity against a variety of cancers (e.g., leukemia, breast, ovarian, prostate, bladder, and colon, as well as melanomas and neuroblastoma) [13]. Several mechanisms of costunolide action have been reported in cancer cells, including cell cycle arrest, metastasis, invasion, apoptosis induction, angiogenesis, and telomerase inhibition [14,15]. Costunolide blocks cell proliferation in skin cancer cells by suppressing ERK, STAT3, and NFκB pathways [16]. Costunolide exerts its anticancer effect by inhibiting the activation of NFκB pathway and nuclear translocation of p50/p65 NFκB subunits in glioma and breast cancer cells [17,18]. Sesquiterpene lactones prevent NFκB activation by alkylating Cys38 in the p65 subunit, and by blocking IκB phosphorylation [19,20]. A variety of signaling pathways, such as AKT phosphorylation, the Wnt-/β-catenin pathway, JNK activation, mTOR, ROS mediated Ras signaling, microtubule assembly, and the STAT3 pathway are inhibited by costunolide [21,22,23,24].

The current preliminary study was designed to measure the mRNA expression of *EpCAM*, a cell proliferation gene, in acute lymphoblastic leukemia. Further, in in vitro experiments, *EpCAM* and its downstream target genes *Myc* and *TERT* were targeted by anticancer agent costunolide in Jurkat cells in order to study its potential therapeutic effect.

## 2. Materials and Methods

### 2.1. Patient Samples

Blood samples of 215 ALL cases and 89 controls were obtained from hospitals with consent from participants. The Helsinki Declaration guidelines were followed. The study was also approved by the Ethics committee of the Department of Biosciences, COMSATS University Islamabad (CUI) and participating hospitals. The participants were grouped into two categories based on age: the pediatric group (≤18 years) and the adult group (>18 years). The cases which had been previously treated for any other carcinoma;which had a medical history of Hep A, B, or C;or who were child-bearing women were excluded from the study. Inclusion criteria included newly diagnosed ALL cases.

### 2.2. RNA Isolation

The whole blood was subjected to RBCs lysis for isolation of the WBCs pellet. RNA was then isolated from WBCs by TRIzol^TM^ reagent (Invitrogen, Waltham, MA, USA). Quantification of RNA was performed by nanodrop, and its integrity was checked on 1% agarose gel. Later, RNA was processed for cDNA synthesis by using MMLV reverse transcriptase (Thermo Fisher, Waltham, MA, USA).

### 2.3. Cell Culture

The Jurkat cell line (ATCC, Manassas, VA, USA) was tested for the presence of any bacterial or fungal contamination before use. The cells were cultured at 37 °C under conditions of 95% air and 5% CO_2_ in RPMI, with 10% FBS and 100X Pen strep solution (Gibco, Grand Island, NY, USA). A stock solution (43 mM) of costunolide (Sigma Aldrich, Burlington, MA, USA) was prepared in 100% DMSO, which was later diluted with RPMI 1640 medium to prepare working concentrations. The final concentration of DMSO was kept below 0.1% to avoid its toxic effect on cells.

### 2.4. Cell Viability Assay

The effect of costunolide on the Jurkat cells’ viability was assessed through MTT assay. The cells were grown in 96-well plates at a density of 2 × 10^4^ cells/well. The control and blank plates were incubated at 37 °C. The assay was performed in triplicate. After 24 h of incubation, costunolide was added in varying concentrations of 9, 8, 7, 6, 5, 4, 3, 2, 1, and 0.5 μM. The plates were again incubated at 37 °C for another 24 h. MTT reagent was added to each well, and after 3 h incubation, purple-colored formazan crystals were dissolved by adding 10% SDS. Using the ELIZA plate reader, cell viability was analyzed by measuring absorbance at 490 nm, and the IC_50_ value of costunolide for Jurkat cells was calculated.

### 2.5. In Vitro Treatment of Jurkat Cells with Costunolide

Jurkat cells were cultured at a density of 0.5 × 10^5^ cells/mL in three 24-well plates, with a final volume of 1 mL/well. Cells without costunolide were also seeded into each plate, and served as a control. All three of the plates were incubated at 37 °C for 24 h. After 24 h, varying concentrations of costunolide (6, 5, 4, 3, and 2 μM) were added into the wells of all the plates except the control, and then incubated for 24, 48, and 72 h as appropriate. RNA and DNA were isolated from costunolide-treated Jurkat cells by using the Qiagen RNeasy^®^ Mini Kit and the DNeasy^®^ blood and tissue kit.

### 2.6. Quantitative Real Time PCR

MMLV reverse transcriptase was used to prepare 500 ng (from blood) and 200 ng (Jurkat cell lines) cDNA from the RNA. cDNA was later used as a template for qPCR. The analysis was performed by using 2X Syber green qPCR master mix on the ABI Step One detection system. The sequence of primers was *EpCAM* (F): 5′-TGCAGGGTCTAAAAGCTGGT-3′, (R): 5′-TGCATCTCACCCATCTCCTT-3′; *Myc* (F): 5′-TCGGATTCTCTGCTCTCCTC-3′, (R): 5′-CCTGCCTCTTTTCCACAGAA-3′; TERT (F): 5′-ATCAGACAGCACTTGAAGAGGGTG-3′, (R): 5′-CCCACGACGTAGTCCATGTTCAC-3′; *NFκB* (F): 5′-TTTCTTCCGGATAGCACTGG-3′, (R): 5′-CCAGCTGTCCTGTCCATTCT-3′; *β-actin* (F): 5′-CTGAACCCCAAGGCCAAC-3′, (R): 5′-AGAGGCGTACAGGGATAGCA-3′; *β-globin* (F): 5′-GCTTCTGACACAACTGTGTTCACTAGC-3′, (R): 5′-CACCAACTTCATCCACGTTCACC-3′. Data were recorded at the extension step and analyzed using the comparative Ct method. β-actin was used as a control for normalizing blood samples, and β-globin was used for normalization of costunolide-treated Jurkat cells.

### 2.7. Measurement of Telomere Length

DNA isolated from control, as well as costunolide-treated Jurkat cells were subjected to measurement of telomere lengthat three different time points, i.e., 24, 48 and 72 h. The concentration of DNA was 35 ng in each 20 μL reaction tube. Telomere length was measured by real-time PCR using the method described by Cawthon [25], with slight modification. The sequence of primers was *TEL* (F): 5′-GGTTTTTGAGGGTGAGGGTGAGGGTGAGGGTGAGGG-3′, (R) 5′-TCCCGACTATCCCTATCCCTATCCCTATCCCTATCCCT-3′, and *β-globin* (as a single copy gene) (F): 5′-GCTTCTGACACACTGTGTTCACTAGC-3′, (R): 5′-CACCAACTTCATCCACGTTCACC-3′, respectively. Telomere length was represented in terms of relative telomere length (T/S) [26].

### 2.8. Docking of Costunolide with EpCAM and Myc Proteins

The protein structures of EpCAM (ID: 4MZV) and Myc (ID: 1NKP) were retrieved from the protein data bank (PDB). The structure of the costunolide ligand was obtained from PubChem (CID 5281437). In silico analysis was performed based on the previous studies of EpCAM interaction with ligands via Ile 170A, Tyr 174A, Phe 216A, Val 220A, Asp 232A, and Leu 242A amino acids [27]. Water molecules, protein, and DNA contents were removed, except for chain A of EpCAM. Similarly, chain A of Myc was selected, and from previously published data regarding potential binding sites of Myc bHLH, the following binding sites were predicted: Arg925, Asp926, Gln927, Ile928, Pro929, Leu931, Glu932, Glu935, Lys936, Ala937, Pro938, Lys939, and Ile942 [28]. In order to predict the interaction of EpCAM and Myc proteins with the ligand costunolide, AutoDock 4.2 was used. Docking runs were set to 50 so that costunolide could bind freely anywhere on EpCAM and Myc, as well as to allow 50 different conformations of EpCAM and Myc to bind with costunolide. Grid size was adjusted to cover entire macromolecules (EpCAM and Myc), in order to find potential binding sites for costunolide. Docking results were further verified by LigPlot^+^. Based on energy values obtained through docking, macromolecule (EpCAM, Myc) binding with costunolide was confirmed.

### 2.9. Computational Analysis

The Eukaryotic Promoter Database [29] was used to locate the binding sites of transcription factors within the promoter regions (1 kb from TSS) of the *EpCAM*, *Myc*, and *TERT* genes. The sequence of *EpCAM* (ENSG00000119888), *Myc* (ENSG00000136997), and *TERT* (ENSG00000164362) promoter regions were retrieved from Ensembl to identify the binding sites of transcription factors.

### 2.10. Statistical Analysis

Student’s *t* test was performed between the control and the cases by using Graph Pad prism software 8 (GraphPad Software 8.4.3, San Diego, CA, USA). One-way ANOVA was applied in order to calculate the statistical significance. The data with *p* values < 0.05 were considered statistically significant.

## 3. Results

### 3.1. Association of EpCAM Gene Expression with Demographic Data

In the pediatric group, there wasa total of 153 cases and 50 control samples, while the adult group comprised 62 cases and 39 control samples (Table 1). *EpCAM* gene expression was determined in both pediatric and adult groups by qPCR. The expression of the *EpCAM* gene was significantly upregulated in pediatric and adult cases (Figure 2A) relative to their control groups. In the pediatric group, *EpCAM* was elevated 4.2-fold (*p* = 0.04), while in the adult group, it showed 5.35-fold (*p* = 0.042) upregulation compared to the respective control groups.

*EpCAM* expression was also studied, with respect to age among the cases (Figure 2B). However, no significant difference was noted between pediatric (≤18 years) and adult (>18 years) groups. The majority of the cases in both groups were males. Among the ALL subtypes, the B-cell ALL was more prevalent than the T-cell ALL. However, there was no change in *EpCAM* expression with respect to sex orALL subtype between the pediatric and adult cases (Figure 2C,D).

### 3.2. Effect of Costunolide on Jurkat Cell Viability

In order to study the effect of the anticancer agent (costunolide) on cell viability, Jurkat cells were used. A cell viability assay was performed to determine the inhibitory effect of varying costunolide concentrations on the growth of Jurkat cells. The inhibitory effect of costunolide on the growth of Jurkat cells was concentration-dependent. At 1 μMcostunolide concentration, only 5.5% growth inhibition was observed; however, at 9 μM concentration, costunolide had a 98.8% inhibitory effect on the growth of Jurkat cells (Figure 3). The IC_50_ value of costunolide for Jurkat cells, calculated by MTT assay, was, therefore, found to be 5 μM.

### 3.3. Effect of Costunolide on Genes (EpCAM, Myc and TERT) Expression and Telomere Length

EpCAM plays an important role in *Myc* regulation, which further controls *TERT* expression. Following treatment of Jurkat cells with different costunolide concentrations (2–6 μM) at 24, 48, and 72 h, mRNA expression of *EpCAM* and its downstream target genes *Myc* and *TERT* were analyzed by qPCR (Figure 4A–C). It was noted that costunolide inhibited the expression of all the genes, *EpCAM*, *Myc*, and *TERT*, at all time intervals in a dose-dependent manner. A significant reduction in *EpCAM*, *Myc*, and *TERT* expression was evident at 5 μM and 6 μM of costunolide, respectively. Costunolide, therefore, significantly reduced *EpCAM*, *Myc*, and *TERT* expression in Jurkat cells.

As *TERT* expression was downregulated by costunolide, telomere length was also measured in the Jurkat cells in the presence of varying costunolide concentrations at 24, 48, and 72 h of treatment (Figure 4D). No significant change in the telomere lengths of Jurkat cells was noted at 24, 48,or72 h of costunolide treatment relative to the control.

### 3.4. In-Silico Interaction of Costunolide with EpCAM and Myc

Given the significant change in *Myc* expression, an in silico analysis was performed. Docking experiments were analyzed on AutoDock 4.2 to identify binding residues of EpCAM and Myc interacting with costunolide. Docking conformations were ranked on the basis of lowest energy. The binding energy determined for EpCAM and Myc was −9.31 kcal/mol and −8.28 kcal/mol, respectively. Negative energy values suggested stable binding of EpCAM and Myc with costunolide. Amino acid residues Ile 170, Tyr 174, and Phe 216 in EpCAM (conformation 1) were involved in hydrophobic interactions with costunolide (Figure 5A). Similarly, Arg 925, Asp 926, Ile 928, Leu 931, and Glu 932 of Myc (conformation 1) were identified to be the potential amino acid residues responsible for its interaction with costunolide (Figure 5B). These results were further confirmed by LigPlot^+^. The similarity of the results with previous studies highly suggests that binding of costunolide with EpCAM and Myc is effective.

### 3.5. Identification of Transcription Factor Binding Sites within EpCAM, Myc, and TERT Promoter Region

The Eukaryotic Promoter Database was used to analyze the promoter region of all the genes for a common transcription factor. We found binding sites of NFκB1/p50 and p65 subunits within the promoter regions of *EpCAM*, *Myc,* and *TERT* genes 1kb from the transcription start site (TSS) (Table 2).

### 3.6. Costunolide Reduced NFκB Expression

NFκB was identified as a common transcription factor of *EpCAM*, *Myc,* and *TERT*; therefore, we further investigated the effect of costunolide on the expression of the *NFκB* gene. The effect of costunolide (2–6 μM) on the transcription of *NFκB* was examined in costunolide-treated Jurkat cells at three time points, i.e., 24, 48, and 72 h, respectively. The expression of *NFκB* was reduced at all concentrations relative to the controls (Figure 6A). *NFκB* expression in Jurkat cells followed a concentration-dependent decrease at 48 and 72 h of costunolide treatment. Thus, costunolide caused significant reduction in *NFκB* expression at all time intervals. Based on the expression data of *NFκB* and its binding sites within the promoter region of all the genes, it was suggested that costunolide could have exerted its inhibitory effect on *EpCAM*, *Myc,* and *TERT* via NFκB, and, therefore, NFκB could act as a regulator of all three genes (Figure 6B).

## 4. Discussion

EpCAM is a tumor-associated antigen, but its role in ALL is not yet well elucidated. EpCAM, along with β-catenin, helps to regulate *Myc,* whose aberrant expression in tumor cells enablesitto bypass the cell cycle, resulting in uncontrolled proliferation [8,30,31]. Elevated expression of *EpCAM* in many cancers makes it an important therapeutic target [32]. Zheng et al. [7] found that induced *EpCAM* expression in acute myeloid leukemia is linked with chemoresistance and activation of the Wnt signaling pathway. In the present study, we found induced *EpCAM* expression in ALL cases as compared to control groups. Over-expression of *EpCAM* has been detected in colon, breast, lung, intestine, and prostate cancers [33]. The elevated expression of *EpCAM* in ALL cases suggests that EpCAM, being a regulator of cell proliferation, has a role in leukemia progression, and that its protein might be a potential target for therapy. However, when *EpCAM* expression was studied in ALL cases with respect to age, sex, and ALL subtypes, no significant difference between pediatric and adult groups was noted. Similar results have been reported in hypopharyngeal and oral squamous cell carcinoma, in which no association has been found between *EpCAM* expression and age and sex [34,35].

Costunolide exhibits significant anticancer properties in different leukemia cell lines, particularly K562, HL-60, THP-1, and Molt-4, either by inducing cell cycle arrest or apoptosis, or by inhibition of NFκB activation [8,11]. In order to investigate the effect of costunolide on *EpCAM* and its downstream target genes, *Myc* and *TERT*, Jurkat cell lines were treated with different costunolide concentrations. *EpCAM*, *Myc,* and *TERT* showed a concentration-dependent decrease at all time intervals. The maximum inhibitory effect of costunolide on the transcriptional activity of *EpCAM*, *Myc,* and *TERT* was noted at 6 μM of the drug. Overall, costunolide significantly reduced *EpCAM*, *Myc,* and *TERT* genes’ expression at 5 μM and 6 μM concentrations, at all time points. In breast cancer cell lines, downregulation of *EpCAM* reduces proliferation, migration, and invasion [36]. Choi et al. [37] observed costunolide-induced differentiation in HL-60 leukemia cells with reduced Myc protein levels. Similarly, costunolide mediated reduced telomerase mRNA and protein levels havealso been reported in human B-cell leukemia [38]. Rapamycin reduced *TERT* expression at the transcriptional level in Jurkat cells by preventing the binding of transcription factors at its promoter region [39].

As costunolide exerted an inhibitory effect on *TERT* expression, the effect of costunolide on the telomere length of Jurkat cells was also explored. It was observed that even 24 to 72 h after costunolide treatment, no change in telomere lengths between the control and costunolide-treated Jurkat cells was noted. In T cells, telomere length didnot change over a shorter time interval, even in the absence of telomerase [40]. Zach et al. [39] reported that in rapamycin-treated Jurkat cells, mild telomere shortening was noticed over the course of 6 months, from 5.1 kb to 3.8kb. Contrastingly, a study has reported telomere shortening in Jurkat cells treated with *Rhodospirillum rubrum* L-asparaginase mutant (RrA) that downregulated telomerase activity [41].

In order to predict the interaction of costunolide with EpCAM and Myc, we performed in silico studies. The amino acid residues of EpCAM (Ile 170, Tyr 174, Phe 216) and Myc (Arg 925, Asp 926, Ile 928, Leu 931, Glu 932) showed hydrophobic interactions with costunolide. Several studies have reported interaction of EpCAM with different ligands and aptamers via amino acid residues Lys 155, Arg 163, Lys 168, Arg 173, Tyr 174, and Lys 221, respectively [1,42,43]. Similarly, the amino acid residues of Myc involved in hydrogen bonding with ligands include Leu 917, Lys 918, Phe 921, Phe 922, Trp 935, Ile 936, Ile 937, and Phe 938 [44,45].

In order to further elucidate the downregulation of selected genes’ expression by costunolide, we analyzed the promoter region of *EpCAM*, *Myc*, and *TERT* genes, using the Eukaryotic Promoter Database to find a common transcription factor. NFκB was identified to be a common transcription factor that regulated the transcription of all these genes. Studies have shown binding sites of transcription factor NFκB within the *EpCAM*, *Myc,* and *TERT* promoter region [46,47,48,49,50,51].

In order to examine the effect of costunolide on *NFκB*, Jurkat cells were treated with varying amounts of costunolide. Costunolide significantly reduced *NFκB* transcription in a concentration-dependent manner. Several studies have reported dose-dependent *NFκB* reduced expression by costunolide [52]. The inhibitory effect of costunolide on the transcriptional activity of *NFκB* and the translocation of p65 from cytoplasm to nucleus was also evident from previous studies [17,53]. Pitchai et al. [54] reported that amino acid residuesof NF-κB, such as Gly238 (NF-κB/p100), Lys153 (NF-κB/p52), Thr191, and Ala192 (NF-κB/p65), interact with costunolide via hydrogen bonds.

## 5. Conclusions

Together, our results suggested that the *EpCAM* gene could be used as a potential cell proliferation marker in Jurkat cells. Costunolide, with antiproliferative activity, interacts with multiple target genes. The drug initially reduced *NFκB* transcription, which might further inhibit downstream targets *EpCAM*, *Myc*, and *TERT*. Costunolide, therefore, could be a promising therapeutic agent whichcan be used to slow Jurkat cell proliferation, due to its inhibitory effect on *EpCAM* and its downstream target genes.

## Figures and Tables

**Figure 1 biomedicines-10-02985-f001:**
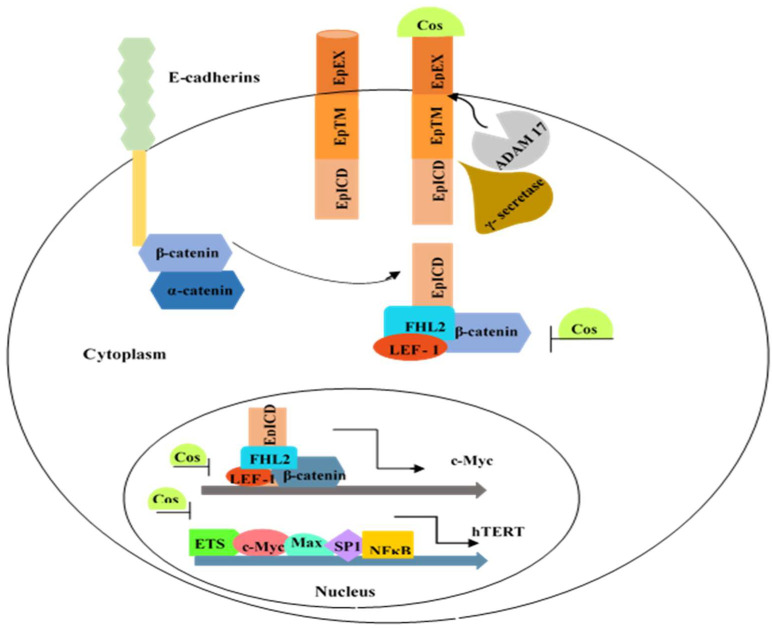
Molecular mechanism of EpCAM and its downstream *Myc* and *TERT* genes. Intracellular domain of EpCAM, cleaved by γ-secretase, associates with β-catenin, LEF-1, and FHL2. This complex translocates to the nucleus and regulates *Myc* transcription. Myc acts as a transcription factor and binds to the *TERT* promoter. EpEX = EpCAM extracellular domain, EpICD = EpCAM intracellular domain, TM = transmembrane domain.

**Figure 2 biomedicines-10-02985-f002:**
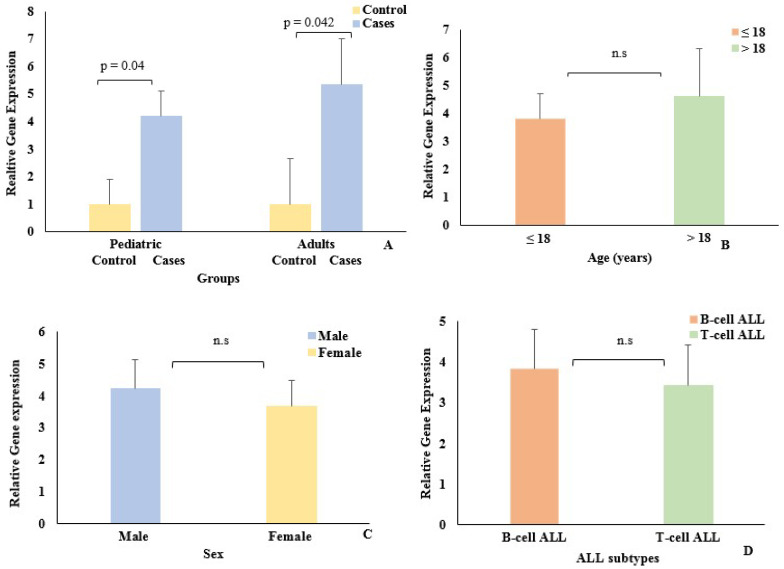
EpCAM expression in (**A**) ALL cases (pediatric and adults) and controls. (**B**) Age group (pediatric ≤ 18 years, adults > 18 years); (**C**) Sex; (**D**) ALL subtype. n.s represents *p* values > 0.05 (*t* test). Error bars represent standard error of mean.

**Figure 3 biomedicines-10-02985-f003:**
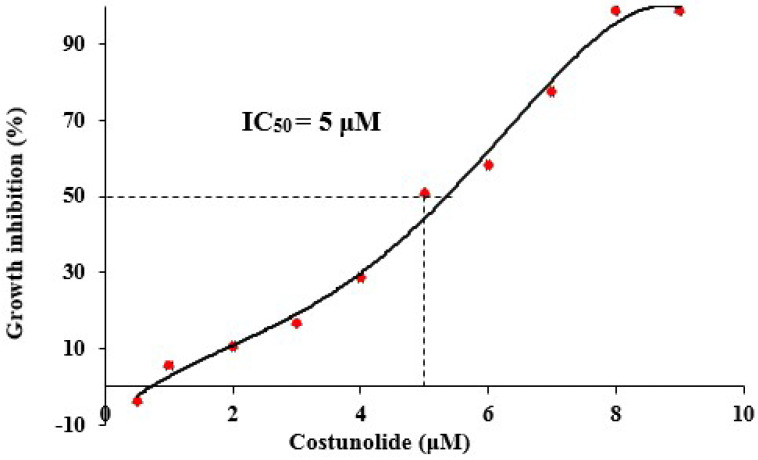
Growth inhibition (%) of Jurkat cells by costunolide and calculation of IC_50_ value. Inhibitory effect of varying costunolide concentrations (0.5–9 μM) on Jurkat cells growth was monitored. Dotted line represents IC_50_ value (5 μM).

**Figure 4 biomedicines-10-02985-f004:**
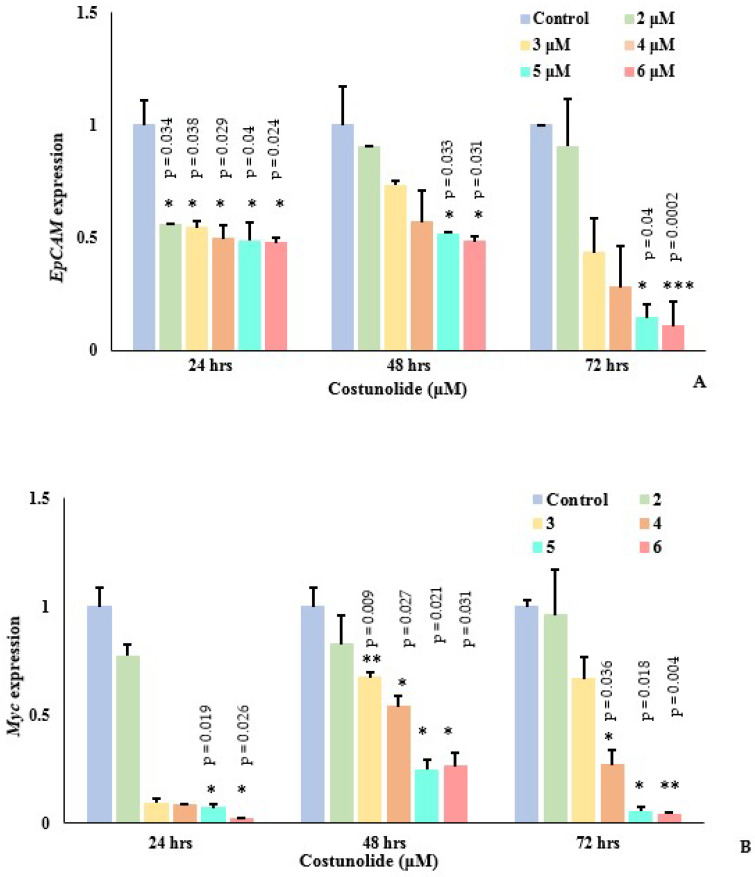
Effect of costunolide on *EpCAM*, *Myc*, *TERT* genes, and telomere length. (**A**–**C**) Effect of costunolide on *EpCAM*, *Myc,* and *TERT* mRNA expression in Jurkat cells at 24, 48, and 72 h. (**D**) Telomere length in costunolide-treated (2–6 μM) and untreated Jurkat cells at three different time points (24, 48, and 72 h). n.s indicates non-significant *p* value > 0.05. *, **, *** indicates *p* values < 0.05, < 0.01 and < 0.001 (one-way ANOVA). Error bars represents standard error of mean. T/S indicates relative telomere length.

**Figure 5 biomedicines-10-02985-f005:**
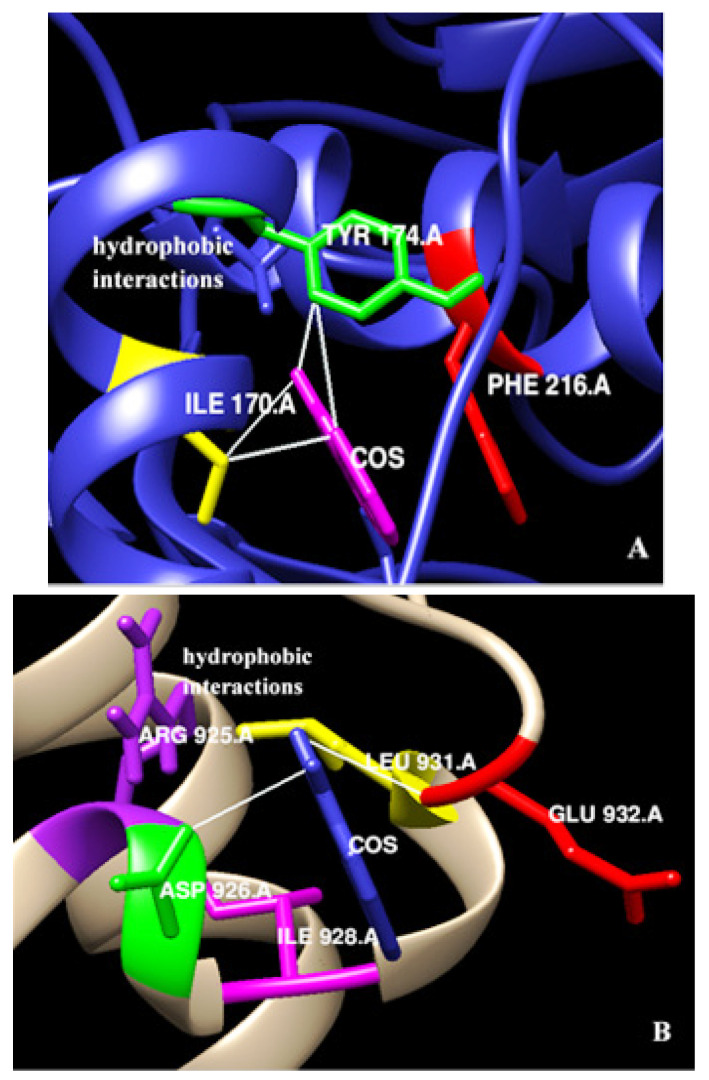
3D representation of the interaction of EpCAM and Myc with costunolide. (**A**) Amino acids Ile170A (yellow), Tyr174A (green) and Phe216A (red) of EpCAM displayed hydrophobic interactions (represented by white lines—few interactions were shown) with costunolide (magenta). (**B**) Amino acids Arg925A (purple), Asp926A (green), Ile928A (magenta), Leu931A (yellow), and Glu932A (red) of Myc displayed hydrophobic interactions (represented by white lines—few were shown) with costunolide (navy blue).

**Figure 6 biomedicines-10-02985-f006:**
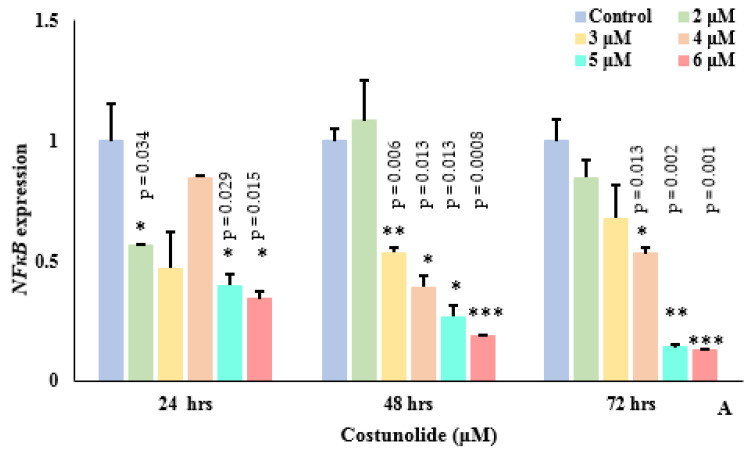
Costunolide and NFκB. (**A**) Effect of costunolide on *NFκB* expression at 24, 48, and 72 h. Error bars represent standard error of mean. *, **, *** indicate *p* values < 0.05, < 0.01, and < 0.001 (One-way ANOVA). (**B**) Inhibitory effect of costunolide on *EpCAM*, *Myc,* and *TERT* via NFκB.

**Table 1 biomedicines-10-02985-t001:** Demographic data of control and cases.

Cases vs. Control						
	Pediatric			Adults		
Characteristics	Controls	Cases	*p* Value	Controls	Cases	*p* Value
**Subjects (N)**	50	153	-	39	62	-
**Age (years)**	2.77 (3.25)	8.58 (4.92)	**0.0026**	30.0 (9.04)	31.3 (9.12)	0.5983
**Sex (Male) (%)**	31 (62.0%)	100 (65.4%)	0.830	31 (79.5%)	49 (79.0%)	0.797
**ALL immunophenotypes**						
**B-cell ALL**117 (76.5%)45 (72.5%)0.3413						
**T-cell ALL**36 (23.5%)17 (27.7%)0.9218						

Values are either N (%) or mean (SD). *p* values are calculated from chi square or Mann–Whitney U test.

**Table 2 biomedicines-10-02985-t002:** Binding sites of NFκB1/p50 and p65 subunits within the promoter regions (1 kb from TSS) of *EpCAM*, *Myc*, and *TERT* genes.

NFκB Subunits	*EpCAM*	*Myc*	*TERT*
**NFκB1/p50**	−916, −691, −134 bps	−161, −7 bps	−742, −727, −694, −692 bps
**p65/RelA**	−393 bps	−259 bps	−599, −598 bps

Bindingsites of NFκB subunits in the promoter regions of *EpCAM*, *Myc*, and *TERT* genes are represented 1 kb upstream to TSS.

## Data Availability

Datas are available upon request from the authors.

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
