# Peer review of "Epithelial Cell Adhesion Molecule (EpCAM) Expression Can Be Modulated via NFκB"

_biomedicines, 2022, doi:10.3390/biomedicines10112985_

Round 1

Reviewer 1 Report

This manuscript describes that EPCAM is regulated by NFKB promoter in Jurkat cell line.

- It would be better to use different colours for the figures instead of different grey patterns to make it easier to the reader to identify the different conditions.

- Figure 2 and 6 quality must be improved.

- Statistics should be described in the figure captions.  I suggest representing the exact p value on the figure instead of an asterisk.

- Results must be validated on other lymphocyte cell lines or primary T cells.  For now, to have "Jurkats" instead of "Leukemia" in the title is more correct.

- Jurkat ATCC reference is missing.

Author Response

Response to the Comments of Reviewer 1:

1) It would be better to use different colors for the figures instead of different grey patterns to make it easier to the reader to identify the different conditions.

All the figures have been updated and thegrey pattern have been replaced by the colored bars in the manuscript.

2) Figure 2 and 6 quality must be improved.

Figure 2 & 6 have been updated with Improved quality.

3) Statistics should be described in the figure captions. I suggest representing the exact p value on the figure instead of an asterisk.

The Figure description has been updated and the statistics have been mentioned.The statisticallysignificant p values have been mentioned in the figures.

4) Results must be validated on other lymphocyte cell lines or primary T cells.  For now, to have "Jurkat" instead of "Leukemia" in the title is more correct.

The word Leukemia has been removed from the title of the manuscript.

5) Jurkat ATCC reference is missing.

ATCC reference for Jurkat Cell line has been added in the materials and methods section.

Reviewer 2 Report

The originality of this paper is not high becuase costunolide is a well reported molecule that suppress the growth of leukemia. In addition, the effect of costunoslide on Nf-KB, Myc and Tert has also been reported. More detailed comments are as follows:

1. The effect of costunolide on the expression of Myc, EpCAM and Tert should be validated at protein level with western blot.

2. All experiments should be done in at least 2 cell lines to exclude cell line specific effects.

3. In silico prediction is very preliminary. Without experimental evidence, it is hard to conclude that costunolide will bind to Myc or EpCAM. 

4. The function of Nf-KB as a transcriptional factor should be validated by ChIP-PCR. 

5. To verify the role of Nf-KB, authors should use Nf-KB blockers or siRNA to block Nf-KB, and then examine target gene expression at mRNA and protein levels. 

Author Response

Response to Reviewer 2

  1. The effect of Costunolide on the expression of Myc, EpCAM and Tert should be validated at protein level with western blot.

Effect of Costunolidein cellular differentiation, inhibitionof cell proliferation, its antiapoptotic and antitelomerase activity has been studied in human myeloid leukemia and promyelocytic leukemia cell lines likes HL-60, K562, U937, Molt-4 [1-7]. Inhibitory effect of Costunolide on Myc has been reported at the protein level in leukemia cell line HL-60 and breast cancer cell lines MCF-7 and MDA-MB-231 [4,8]. Similarly, Costunolide has also been found to reduce the protein expression of TERT in NALM-6 cells [9]. The exact mechanistic details, however are not known so far. To the best of our knowledge this is the first report on the possible mechanism of transcription regulation of EpCAM. It is also the first report on transcription regulation of Cmyc and Tert through NFkBby Costunolide. As rightly pointed out by the reviewer western blot would have validated the results but due to lack of funding, we couldn’t do protein analysis.

  1. All experiments should be done in at least 2 cell lines to exclude cell line specific effects.

Effect of Costunolide in cellular differentiation, inhibition of cell proliferation, its antiapoptotic and antitelomerase activity has been studied in human myeloid leukemia and promyelocytic leukemia cell lines likes HL-60, K562, U937, Molt-4 at protein level before[1-7]. The current report is a pilot study focusing on the possible mechanistic detail of the mode of action of Costunolide. The experiments done in more than one cell line would have strengthen the hypothesis, but lack of funds restricted us to one cell line.As per suggestion of the first reviewer, we have removed the word “leukemia” from the title to specify that the effect of the costunolide on the selected genes were studied in Jurkat cells only. 

  1. 3. In silico prediction is very preliminary. Without experimental evidence, it is hard to conclude that costunolide will bind to Myc or EpCAM.

The reviewer has rightly pointed out that it’s hard to conclude that Costunolide will bind to Myc or EpCAM just based on in silico analysis. We hypothesized in this paper that if Costunolide does inhibit NFkB [15-16] and is anexperimentally verified transcription factor of EpCAM [10], Myc and TERT genes [10-14]. Then treating cells with Costunolide should also inhibit the transcription of EpCAM, Myc and TERT (Figure B). Our transcription data validate the above mention hypothesis and is first report that Costunolide can have a therapeutic potential by inhibiting EpCAM transcription via NFκBinhibition.  This was the missing link in the literature was that if Costunolide inhibits NFkB does it also effect the its downstream targets EpCAM, Myc and TERT? This manuscript provides a preliminary answer to this question that Costunolide can inhibit NFkB and its downstream targets.

(B) Inhibitory effect of costunolide on EpCAM, Myc and TERT via NFκB

  1. The function of NFkB as a transcriptional factor should be validated by ChIP-PCR.

There are experimental evidences reported in literature for NFkBas transcription factor ofEpCAM, Myc and TERT genes [10-14]. The ChIP analysis had been carried out to confirm NFkB as transcription factor ofEpcam [10]. While it was validated through Western blot and EMSA thatNFkB is a transcription factor forCmyc and Tertgenes [12,13]. These literature reports provide clear experimental proof that NFKB is a transcription factor of Myc,Tertand EpCAM.

  1. To verify the role of NFkB, authors should use NFkB blockers or siRNA to block NFkB, and then examine target gene expression at mRNA and protein levels.

In this paper we provide the proof of concept that Costunolide, a reported NFkBinhibitor/blocker, can have therapeutic potential by inhibiting Epcam [15-16]. TreatingJurkat cells with Costunolide had inhibitory effect on NFkB at mRNA level. It will be great if the similar results can be validated using some other siRNA or NFkB blockersto compare the inhibitory effects but at the moment we have funding constraint to perform these experiments.

References:

1) Lee, M. G., Lee, K. T., Chi, S. G., & PARK, J. H. (2001). Constunolide Induces Apoptosis by ROS-mediated Mitochondrial Permeability Transition and Cytochrome C Release. Biological and Pharmaceutical Bulletin, 24(3), 303-306.

2) Kim, S. H., Kang, S. N., Kim, H. J., & Kim, T. S. (2002). Potentiation of 1, 25-dihydroxyvitamin D3-induced differentiation of human promyelocytic leukemia cells into monocytes by costunolide, a germacranolide sesquiterpene lactone. Biochemical pharmacology, 64(8), 1233-1242.

3) Saosathan, S., Khounvong, J., Rungrojsakul, M., Katekunlaphan, T., Tima, S., Chiampanichayakul, S., ... &Anuchapreeda, S. (2021). Costunolide and parthenolide from ChampiSirindhorn (Magnolia sirindhorniae) inhibit leukemic cell proliferation in K562 and molt-4 cell lines. Natural Product Research, 35(6), 988-992.

4) Choi, J. H., Seo, B. R., Seo, S. H., Lee, K. T., Park, J. H., Park, H. J., ... & Miyamoto, K. I. (2002). Costunolide induces differentiation of human leukemia HL-60 cells. Archives of pharmacal research, 25(4), 480-484.

5) Choi, J. H., Ha, J., Park, J. H., Lee, J. Y., Lee, Y. S., Park, H. J., ... & Lee, K. T. (2002). Costunolide triggers apoptosis in human leukemia U937 cells by depleting intracellular thiols. Japanese Journal of Cancer Research, 93(12), 1327-1333.

6) Cai, H., He, X., & Yang, C. (2018). Costunolide promotes imatinib‐induced apoptosis in chronic myeloid leukemia cells via the Bcr/Abl–Stat5 pathway. Phytotherapy Research, 32(9), 1764-1769.

7) Jiang, J., Cai, H., Qu, J., & Yang, C. H. (2020). Costunolide suppresses proliferation of K562 cells through JAK/STAT signaling pathway. Zhongguoshiyanxue ye xuezazhi, 28(2), 460-463.

8) Peng, Z., Wang, Y., Fan, J., Lin, X., Liu, C., Xu, Y., ... & Su, C. (2017). Costunolide and dehydrocostuslactone combination treatment inhibit breast cancer by inducing cell cycle arrest and apoptosis through c-Myc/p53 and AKT/14-3-3 pathway. Scientific reports, 7(1), 1-16.

9) Kanno, S. I., Kitajima, Y., Kakuta, M., Osanai, Y., Kurauchi, K., Ujibe, M., & Ishikawa, M. (2008). Costunolide-induced apoptosis is caused by receptor-mediated pathway and inhibition of telomerase activity in NALM-6 cells. Biological and Pharmaceutical Bulletin, 31(5), 1024-1028.

10) van der Gun, B.T.; de Groote, M.L.; Kazemier, H.G.; Arendzen, A. J.; et al. Transcription factors and molecular epigenetic marks underlying EpCAM overexpression in ovarian cancer. British journal of cancer, 2011, 105(2), 312-319.

11) Herreros-Pomares, A.; Aguilar-Gallardo, C.; Calabuig-Fariñas, S.; Sirera, R.; Jantus-Lewintre, E.; Camps, C. EpCAM duality becomes this molecule in a new Dr. Jekyll and Mr. Hyde tale. Critical reviews in oncology/hematology. 2018, 126, 52-63.

12) Huang, H.; Ma, L.; Li, J.; Yu, Y.; et al. NF-κB1 inhibits c-Myc protein degradation through suppression of FBW7 expression. Oncotarget. 2014, 5(2), 493.

13) Yin, L.; Hubbard, A. K. Giardina, C. NF-κB regulates transcription of the mouse telomerase catalytic subunit. J Biol Chem. 2000, 275(47), 36671-36675.

14) NF-kB Target Genes. https://www.bu.edu/nf-kb/gene-resources/target-genes/. Accessed 10th November 2022.

15) Choi, Y. K., Cho, S. G., Woo, S. M., Yun, Y. J., Jo, J., Kim, W., ... & Ko, S. G. (2013). Saussurealappa Clarke-derived costunolide prevents TNFα-induced breast cancer cell migration and invasion by inhibiting NF-κB activity. Evidence-Based Complementary and Alternative Medicine, 2013.

16) Liu, B., Rong, Y., Sun, D., Li, W., Chen, H., Cao, B., & Wang, T. (2019). Costunolide inhibits pulmonary fibrosis via regulating NF-kB and TGF-β1/Smad2/Nrf2-NOX4 signaling pathways. Biochemical and biophysical research communications, 510(2), 329-333.

Round 2

Reviewer 2 Report

The authors have addressed the comments in part.